# Thermalization, Ergodicity and Quantum Fisher Information

César Gómez

*Instituto de Física Teórica UAM-CSIC, Universidad Autónoma de Madrid, Cantoblanco, 28049 Madrid, Spain*

The eigenstate thermalization hypothesis as well as the quantum ergodic theorem are studied in the light of quantum Fisher information. We show how global bounds on quantum Fisher information set the ETH and ergodicity conditions. Complexity and operator growth are briefly discussed in this frame.

A fundamental problem in physics is to understand why statistical methods, based on incomplete (macroscopic) knowledge, are most of the time very accurate when we consider systems with large number of degrees of freedom. The essence of this problem is the quantum mechanics version of the ergodic hypothesis as it was first addressed by von Neumann in 1929 [1].[1] In a nutshell the problem lies in understanding the relation between time averages and microcanical ensembles.

In its most modest version, as it was addressed in the 1920's, the problem can be presented as follows. Let us consider a many body dynamical system with large number $N$ of degrees of freedom and with a Hilbert space $\mathcal{H}$ of finite but large dimension $\mathcal{D}$. Next you assume that the exact microscopic Hamiltonian $H$ has energy eigenstates $|\alpha\rangle$ with $\alpha = 1...\mathcal{D}$ with no degeneracies as well as not resonances i.e. the values of $E_\alpha - E_\beta$ are also non degenerate. In these conditions take a typical initial quantum state $|\psi\rangle$ and define the time average density matrix

$$\rho(|\psi\rangle) \equiv \lim_{T\to\infty} \frac{1}{T} \int_0^T |\psi(t)\rangle\langle\psi(t)| dt \,. \tag{1}$$

The microcanonical ensemble for this Hamiltonian is given by

$$\rho_{\mathrm{mc}} = \sum_\alpha \frac{1}{\mathcal{D}} |\alpha\rangle\langle\alpha| \,. \tag{2}$$

Thus, the question is how close is, for a given Hamiltonian $H$, a given state $|\psi\rangle$ and a given Hilbert space $\mathcal{H}$, the time average $\rho(|\psi\rangle)$ to the microcanonical ensamble $\rho_{\mathrm{mc}}$.

It is a trivial exercise to show that in the absence of energy degenerations for a generic state $|\psi\rangle = \sum c_\alpha |\alpha\rangle$ the time average is given by

$$\rho(|\psi\rangle) = \sum_\alpha c_\alpha c_\alpha^* |\alpha\rangle\langle\alpha| \,. \tag{3}$$

Thus, the question about how close is the time average from the microcanical ensemble should be defined, relative to a given observable $A$, as the difference between $\mathrm{Tr}(\rho A)$ and $\mathrm{Tr}(\rho_{\mathrm{mc}} A)$ for a generic state $|\psi\rangle$. In this sense time average is equivalent to the micro canonical ensemble if for a generic $|\psi\rangle$ we can find a set of self adjoint operators for which $\mathrm{Tr}(\rho A) \sim \mathrm{Tr}(\rho_{\mathrm{mc}} A)$ such that we can define a good macroscopic description of the system in terms of these set of operators.

The notion of quantum ergodicity worked out by von Neumann precisely identifies what sort of macroscopic (coarse grained) observable $A^{\mathrm{cg}}$ cannot distinguish between the time average and the microcanonical ensemble.

A simplest question is to identify under what conditions a self adjoint operator $A$ with $[H, A] \neq 0$ satisfies

$$\mathrm{Tr}(\rho(|\psi\rangle) A) \sim \mathrm{Tr}(\rho_{\mathrm{mc}} A) \,, \tag{4}$$

for $H$ a Hamiltonian without degeneracies. If we simply impose (4) we only need to require a simple condition on the diagonal elements $A_{\alpha,\alpha}$ of the operator $A$, namely

$$A_{\alpha,\alpha} = \mathcal{A} = \frac{\mathrm{Tr}(A)}{\mathcal{D}} \,. \tag{5}$$

This leaves completely undetermined the off diagonal elements $A_{\alpha,\beta}$ of the operator $A$. The conditions on the off diagonal elements are obtained if we compare the standard deviations for time averages and the microcanonical ensemble.

Let us introduce the notation

$$< f > = \lim_{T\to\infty} \frac{1}{T} \int_0^T f(t)\, dt \,, \tag{6}$$

for the time averages. For a given observable $A$ and a generic state $|\psi\rangle$ let us define the vN function [2]

$$G(A, |\psi\rangle) = < (A_\rho - < A_\rho >)^2 > = < A_\rho^2 > - (< A_\rho >)^2 \,, \tag{7}$$

with $A_\rho = \mathrm{Tr}(\rho(|\psi(t)\rangle) A)$ for $|\psi(t)\rangle$ the time dependent state defined by the hamiltonian evolution $H$. It is a simple exercise to show that the vN function $G$ depends on the off diagonal elements $A_{\alpha,\beta}$ of the operator $A$. Let us assume that we impose the condition (5) on the diagonal elements. The vN quantum ergodicity condition can be now expressed as

$$G(A, |\psi\rangle) \leq \frac{1}{\mathcal{D}^2} \,, \tag{8}$$

for a generic state $|\psi\rangle$. If this condition is satisfied we can say that relative to the observable $A$, the time average and the microcanonical average are equivalent.

---

[1] For an excellent analysis of von Neumann theorem see [2] and references therein.

In reality the bound on $G$ depends on two parameters $\epsilon$ and $\delta$ as follows. For any time we can bound the difference $(A_\rho - <A_\rho>)$ by some epsilon and to require that this difference between the time average and the microcanonical average is bigger than $\epsilon$ only in a small fraction order $\delta$ of the total time relative to some appropriated measure. This will introduce a prefactor in the r.h.s of (8) of order $\epsilon^2\delta$. In our analysis we shall ignore this prefactor setting the ergodicity condition by the exponential suppression $\frac{1}{\mathcal{D}^2}$.

It is easy to see that condition (8) leads to the extra condition on the off diagonal elements

$$A_{\alpha,\beta} \sim \frac{1}{\mathcal{D}}\,. \tag{9}$$

This condition together with (5) is equivalent (see [5]) to the eigenstate thermalization hypothesis (ETH) [3, 4]. Note that at this point of the discussion we are considering the observable $A$ and not any coarse grained version. Moreover, also note that what the vN function $G$ measures is the lack of commutativity $[H, A]$.

In the previous discussion the specific state $|\psi\rangle$ i.e. the concrete values of $c_\alpha$ are not playing a real role after time averaging due to the condition of no degeneracies and no resonances. In this sense the ergodicity condition or equivalently the ETH is given as a set of conditions that depend only on the Hamiltonian $H$ and the operator $A$.

What we will do next is to try to understand what is the quantum information meaning of the vN ergodicity condition. For the operator $A$ we shall define the associated density matrix $\rho_A$ by

$$\rho_A = \frac{1}{\text{Tr}(A)} \sum A_{\alpha,\beta} |\alpha\rangle\langle\beta|\,. \tag{10}$$

Given this density matrix we shall define the quantum Fisher information function (see [6])

$$F(A, H) = \text{Tr}(\rho_A L_A^2)\,, \tag{11}$$

with $L_A$ defined by solving the time evolution equation

$$\frac{\mathrm{d}\rho_A}{\mathrm{d}t} = \frac{L_A\rho_A + \rho_A L_A}{2}\,. \tag{12}$$

We now claim that the vN function for a generic state $|\psi\rangle$ is bounded by $F(A, H)$ i.e.

$$G(A, |\psi\rangle) \leq F(A, H)\,. \tag{13}$$

Hence we will try to justify the following claim:

**Claim:** *For a given Hamiltonian $H$ and a given observable $A$ with $[H, A] \neq 0$ vN ergodicity condition (as well as the ETH hypothesis) is achieved if the quantum Fisher information $F(A, H)$ saturates the statistical Crammer-Rao inequality*

$$F \geq \frac{1}{e^{2S}}\,, \tag{14}$$

*for an entropy $S$ defined as $\ln \mathcal{D}$.*

To fix ideas let us consider a many body system with $N$ degrees of freedom and with a given macroscopic value of the energy $E$. The Hilbert space will be defined by the set of states in the energy interval $[E + \delta(E)/2, E - \delta(E)/2]$. We will assume that this Hilbert space has large dimension $\mathcal{D}$ so the entropy can be defined as $S = \ln \mathcal{D}$.

The Lyapunov equation (12) can be formally solved for $L_A$ as

$$L_A = \int_0^\infty \mathrm{d}\tau e^{-A\tau}[H, A]e^{-A\tau}\,. \tag{15}$$

Defining

$$[H, A](\tau) = e^{-A\tau}[H, A]e^{-A\tau}\,, \tag{16}$$

the solution is given by

$$[H, A](\tau) = [H, A](0)e^{-\frac{\tau[H,A](0)}{L}}\,. \tag{17}$$

In the eigenbasis of $H$ we get

$$L_{\alpha,\beta} = \frac{(\langle\alpha|[H, A]|\beta\rangle)^2}{\langle\alpha|[H, A^2]|\beta\rangle}\,. \tag{18}$$

In terms of the energy eigenvalues

$$L_{\alpha,\beta} = \frac{(E_\alpha - E_\beta)A_{\alpha,\beta}^2}{(A^2)_{\alpha,\beta}}\,. \tag{19}$$

The associated quantum Fisher information, which is constant for a Hamiltonian evolution, is defined by (11).

In order to estimate $F$ we shall make several assumptions on the energy spectrum. In particular we will take $(E_\alpha - E_\beta)$ of the order $\epsilon(\beta - \alpha)$ for $\alpha$ going from 1 to $\mathcal{D}$ with $\epsilon = \frac{\delta(E)}{\mathcal{D}}$ for $\delta(E)$ defined above. This condition implies that we don't have resonances. If in addition we assume that all the off diagonal terms are of the same order let us say $A_{\alpha,\beta} = \mathcal{B}r_{\alpha,\beta}$ with $r$ order one we will get

$$F \sim \frac{\mathcal{B}\delta(E)^2}{\mathcal{A}e^N}\,. \tag{20}$$

The dimensions of $F$ come from using for $L_A$ an operator with formal dimensions of energy. We will define a dimensionless Fisher function as $\mathcal{F} = \frac{F}{\delta(E)^2}$.

In order to define a lower bound to $\mathcal{F}$ we will use the following statistical argument [7]. Recall that for a gaussian probability distribution with variance $\sigma$ the Fisher function defined taking the variance as the parameter is given simply by $\frac{1}{\sigma^2}$. Moreover in this case the standard Shanon entropy satisfies $e^{2S} = \sigma^2$ which leads to the bound

$$\mathcal{F} \geq \frac{1}{e^{2S}}\,, \tag{21}$$

with equality for the gaussian distribution. We shall assume (21) as the Crammer-Rao statistical bound. By saturating this bound we get

$$\mathcal{B} \sim \frac{1}{\mathcal{D}}, \qquad (22)$$

which agrees with the ergodicity and ETH condition (9). In summary we conclude that *ergodicity as well as the ETH hypothesis are achieved when the quantum Fisher information reaches its absolute minimum.*

### A. On a generalized notion of "Rindler time"

A key property of Fisher quantum information is to set, through the Crammer-Rao theorem, the variance of time estimation. In particular minimal Fisher information leads to a maximal variance for the time estimator. As discussed in [8] the time estimator operator is defined, for a given Lyapunov operator $L_A$, by $\frac{L_A}{F(A,H)}$ and the variance of this time estimator is given by $\frac{1}{F(A,H)}$ i.e.

$$\Delta^2(s) \sim \frac{1}{F}, \qquad (23)$$

with $s$ denoting the time estimator. In order to define a dimensionless time estimator we shall replace $F$ by the dimensionless $\mathcal{F}$ introduced above.

Let us now consider the many body system with Hilbert space defined for the set of states in the energy interval around $E$ with width $\delta(E)$. In this case the dimensionless time estimator is defined as

$$\tau = s\delta(E), \qquad (24)$$

with $s$ representing the *physical* time. Note that if the system thermalizes we have $E \sim NT$ for $T$ the temperature. In this case $\delta(E)$ should be of the order $T$ and the dimensionless time $\tau$ defined above becomes the analog of Rindler time.[2]

To see the potential interest of this formal comment let us come back to our previous discussion. We have observed that ergodicity is achieved in the limit of minimal Fisher information given by $e^{-2S}$ that leads to a variance on the corresponding so defined Rindler time of the order

$$e^S, \qquad (25)$$

which is the order of magnitude we expect for the time needed to create maximal complexity [9].

The connection between ETH hypothesis and quantum chaos suggests that *for chaotic Hamiltonians we get minimal Fisher information and that the complexity time $e^S$ is simply the inverse [8] of this minimal Fisher information.*

### B. The effect of coarse graining

In [1] ( see also [2]) the notion of a macroscopic observable was introduced in terms of an orthogonal decomposition of the Hilbert space $\mathcal{H}$ into a set of orthogonal subspaces $\mathcal{H}_\nu$ of dimension $d_\nu$, where each subspace corresponds to states having the same macroscopic value of the observable $A$. The corresponding coarse grained operator can be defined as

$$A^{\mathrm{cg}} = \sum_\nu \rho_\nu P_\nu, \qquad (26)$$

for $P_\nu$ the projector on $\mathcal{H}_\nu$. We can also introduce the value $\mathcal{N}$ as the number of subspaces. The von Neuman function $G(A, |\psi\rangle)$ can be now defined for the coarse grained operator $A^{\mathrm{cg}}$ without mayor changes. The ergodicity bound (8) is then modified to:

$$G \leq \sum_\nu \frac{d_\nu^2}{\mathcal{D}^2 \mathcal{N}}. \qquad (27)$$

The relation with the quantum Fisher information goes as before just modifying the operator $A$ by $A^{\mathrm{cg}}$. The Fisher function $F(A^{\mathrm{cg}}, H)$ is determined by the amplitudes

$$\langle \nu | H | \nu' \rangle, \qquad (28)$$

between different subspaces $\mathcal{H}_\nu$ i.e. by the *macroscopic transition amplitudes.* This modifies the statistical lower bound of $F$. Qualitatively for the simple example with all subspaces of the same dimension $d_\nu$ and with $\mathcal{N}$ subspaces we should get

$$F(A^{\mathrm{cg}}, H) \geq \frac{1}{\mathcal{N}^2}, \qquad (29)$$

corresponding formally to replace the entropy $S = \ln \mathcal{D}$ by $\tilde{S} = \ln \mathcal{N}$.

This coarse grained Fisher function leads, using the same arguments that above, to a dimensionless time scale

$$\tau \sim \mathcal{N}, \qquad (30)$$

that we can interpret as the *macroscopic ergodicity time* relative to the given coarse graining.

It is instructive to consider as a toy model example a case similar to what we expect for a black hole interpreted as a many body system with a Hilbert space of dimension $\mathcal{D} = 2^N$ for $N$ the entropy $S$ and to divide the Hilbert space into $\mathcal{N} = N$ sectors. In this case *the microscopic ergodicity time*, to be defined as the inverse of the minimal Fisher function, is given by the complexity time $e^N$ while the *macroscopic ergodicity time* is given by $N$.[3]

---

[2] For the connection of computational time and Rindler time see [9] and references therein.

[3] Which is the life time of the black hole. Notice we are using dimensionless Rindler time.

### C. Macroscopic complexity

The notion of operator complexity is generically introduced using a particular decomposition of the Hilbert space $\mathcal{H}$ into different subspaces $\mathcal{H}_n$ where here $n$ measures the complexity of states [10]. This can be defined in many ways depending on the physics target, for instance as the number of degrees of freedom contributing to $\mathcal{H}_n$. Let us define a *macroscopic* operator $\mathcal{C}$ as $\sum_n n P_n$ for $P_n$ the projector on $\mathcal{H}_n$. The growth of complexity with time is determined by the *macroscopic amplitudes* $H_{n,m}$ induced by the Hamiltonian. As discussed in [8] the *rate of growth* is determined by the quantum Fisher function

$$F(\mathcal{C}, H). \tag{31}$$

For chaotic systems we have argued that the ergodicity condition implies that $F$ reaches its absolute minimum. This leads to a Lyapunov exponent defined relative to the *macroscopic decomposition* as

$$\lambda \sim \frac{\delta(E)}{\mathcal{N}}, \tag{32}$$

for $\mathcal{N}$ the number of macroscopic subspaces. Recall we are considering a finite dimensional Hilbert space. The former expression leads to a natural upper bound with $\mathcal{N} \sim 1$ corresponding to have a subspace of size order $\mathcal{D}$ where the system equilibrates. This upper bound is in nice agreement with the one suggested using gravity arguments in [11].

It is amusing to define, using the lower bound on the quantum Fisher information, the different time scales associated with different *macroscopic decompositions.* The dimensionless "Rindler" time scale for a given macroscopic decomposition of $\mathcal{H}$ goes as $\mathcal{N}$. In the microscopic case corresponding to $\mathcal{N} = \mathcal{D}$ we get the expected *complexity time* while in the case $\mathcal{N} \sim \ln(\ln(\mathcal{D}))$ corresponding to equilibrium, we get the scrambling time. In summary macroscopic ergodicity time depends on the actual macroscopic decomposition of the Hilbert space [4]. What we know as scrambling corresponds to a macroscopic decomposition where one of the subspaces has essentially the dimension of the full Hilbert space.

**Acknowledgements.** This work was supported by the grants SEV-2016-0597, FPA2015-65480-P and PGC2018-095976-B-C21.

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

[4] The matrix relating the energy eigenbasis and the basis of the macroscopic subspaces is a Haar distributed unitary matrix. In the case $\mathcal{N} \sim 1$ this matrix plays the role of a quantum randomizer.