# Peer review of "Thermalization, Ergodicity and Quantum Fisher Information"

_SciPost Physics_

## Round 1 · Referee Report · Anonymous (Referee 1) · 2020-4-8

Report

The manuscript “Thermalization, Ergodicity and Quantum Fisher Information” by C. Gomez proposes the quantum Fisher information as a tool to study the eigenstate thermalization hypothesis and the quantum ergodic theorem. This is a very interesting and timely topic.

I must admit that I find the paper not very well written, or written too quickly, which makes it not easily accessible to the reader. Admittedly, I got very confused: not all symbols are defined, or their definition appears later than their use, key mathematical steps are not derived and even left as unproved claims [I am talking about the key inequality (13)]. The manuscript should be re-edited and expanded in order to improve the writing and presentation.

  • For instance, Eq.(3) might appear after Eq.(2), or there is some distinction between “typical initial quantum state |psi>” of Eq.(1) and “generic state |psi>” of Eq.(3)? Related to this question: in Eq.(5) I get Tr[rho(psi)A] = |c_alpha|^2*Tr[A]/D and Tr[rho_mc A] = Tr[A]/D^2, and I thus do not recover Eq.(4). I assume that |c_alpha|^2=1/D is an additional condition for Eq.(4) that should be specified.

  • The author should explain how Eq.(8) is derived. Instead of saying “It is a simple exercise to show that the vN function G depends on the off diagonal elements Aα,β of the operator A.” it is maybe better to drop a couple of lines for the exact derivation: I got very confused by the notation.

  • Eq.(8) looks strange to me: If Eq.(4) is satisfied by |c_alpha|^2=1/D, it means that the state is uniformly spread over the whole space and its variance, Eq.(7), should be large.

Surely the author see that I got very confused from reading the introduction: I must certainly improve it and explain all statement in more details. Citations are missing, I think.

  • Why Eq.(10) is called a density matrix. To me it looks like the definition of operator A normalised to its trace, but while the trace of a density matrix is equal to 1 by definition, the trace of a generic operator A can be equal to zero. Equation (10) looks strange. Certainly a crucial assumption is that Tr[A] \neq 0. Please explain.

  • Equation (13) is absolutely non-trivial to me: one the left side there is the state \psi and the operator A, on the right side the operator A (in the form of a density matrix) and the Hamiltonian. Where do Eq.(13) comes from? I did not find the proof anywhere in the text.

  • Crammer -> Cramér

  • Even with all the assumptions, it is not clear to me how Eq.(20) is derived. Where do exp(N) in the denominator comes from ? What is N ? I did not find it anywhere in the text.

  • The claim below Eq.(20) is based on a statistical argument that is valid for the classical Fisher information, but it is used here for the quantum Fisher ifnormation. This is not so straightforward and certainly more explanation is needed. My understanding is that a specific measurement observable “with gaussian probability distribution” is used here, then the classical Fisher is 1/sigma^2 with respect to that measurement observable (which is a specific one). With this, I understand Eq.(14). However, it is not the absolute minimum: it is not guaranteed that I can choose a different observable such that the classical Fisher information is smaller than 1/sigma^2. Maybe the author is considering a natural observable, but explanation is needed.

The highlighted text in italic looks like the main message of this manuscript. However, it is based on the use of Eq.(8) [that is not well derived], Eq.(13) [that really central here but is not derived at all], Eq.(20) [which is also obscure], Eq.(21) [which holds for a specific observable, whose choice and role is unclear here]. I doubt that we can talk about the absolute minimum of the quantum Fisher information here.

The rest of the paper is a collection of sparse thought (I gave up at this point). Also there, intermediate steps and details are not definite.

  • ETH in the abstract is not definite

  • Reference list. Certainly Brenes et al PRL 124, 040605 (2020) is an important reference here. Also, I think it is worth recalling the relation between quantum Fisher information and entanglement, which might be relevant in this discussion, see PRL 102, 100401 (2009) and PRA 85, 022321 (2012) and the reviews RMP 90, 035005 (2018) and J. Phys. A 47, 424006 (2014) which are more up to date than Ref. [6].

In conclusion, I got very interested by the paper and I read it carefully. Unfortunately the paper is not clear and detailed. I got very confused and I did not learned much. As it is, the manuscript is not accessible to any reader. I cannot recommend the current version even though I will be willing to reconsider a more detailed and expanded version, but I will reject a version with incremental improvements.

  • validity: low
  • significance: good
  • originality: high
  • clarity: poor
  • formatting: below threshold
  • grammar: good

Author:  Cesar Gomez  on 2020-04-17  [id 797]

(in reply to Report 1 on 2020-04-08)

First of all thanks for the detailed and constructive report. I agree with most of the criticisms in particular with the complain by the referee about the clarity of the presentation. I think the paper should be certainly improved. Before doing that I would like to answer the concrete comments raised in the report.

For instance, Eq.(3) might appear after Eq.(2), or there is some distinction between “typical initial quantum state $|\psi>$” of Eq.(1) and “generic state $|\psi>$” of Eq.(3)? Related to this question: in Eq.(5) I get $Tr[\rho(\psi)A] = |c_\alpha|^2*Tr[A]/D$ and $Tr[\rho_mc A] = Tr[A]/D^2$, and I thus do not recover Eq.(4). I assume that $|c_\alpha|^2=1/D$ is an additional condition for Eq.(4) that should be specified.

This is correct.

The author should explain how Eq.(8) is derived. Instead of saying “It is a simple exercise to show that the vN function G depends on the off diagonal elements $A_{\alpha,\beta}$ of the operator A.” it is maybe better to drop a couple of lines for the exact derivation: I got very confused by the notation.

This equation is a direct consequence of lemma 4.3 of reference 2 for the particular case of $d_{\nu}=1$. The parameters $\epsilon$ and $\delta$ are briefly discussed in the next paragraph. If needed this equation can be explained in more detail.

Why Eq.(10) is called a density matrix. To me it looks like the definition of operator A normalised to its trace, but while the trace of a density matrix is equal to 1 by definition, the trace of a generic operator A can be equal to zero. Equation (10) looks strange. Certainly a crucial assumption is that $Tr[A] \neq 0$. Please explain.

Here the idea is simply to associate formally the operator $A$ with a density matrix operator. For that the referee is right that the condition $Tr[A] \neq 0$ is assumed.

Equation (13) is absolutely non-trivial to me: one the left side there is the state $\psi$ and the operator A, on the right side the operator A (in the form of a density matrix) and the Hamiltonian. Where do Eq.(13) comes from? I did not find the proof anywhere in the text.

Equation (13) is just a way to introduce the conjecture of the paper. Let me try to explain the philosophy at this point a bit more clearly. The ETH can be thought as follows. Given the operator $A$ and a Hamiltonian $H$ the ETH implies a set of conditions on $A_{\alpha,\alpha}$ and $A_{\alpha,\beta}$ that are equivalent to ergodicity condition as discussed in reference 5. To make this relation is crucial to use Lemma 4.3 of reference 2. In essence the whole point is to discover some conditions relating the basis of $A$ and the one of the Hamiltonian $H$. The comment I try to make is simply the following. Let me associate a density matrix with $A$ in case $Tr(A)$ is finite. Now $H$ defines the time evolution of this density matrix and I can compute the corresponding quantum Fisher information. The representation of this Fisher function in the basis of eigenvectors of $A$ is very simple, namely

$$ F = \sum_{m\neq n} f(m,n) H_{m,n}^2 $$
where $f(m,n)$ depends on the eigenvalues of $A$. In other words the Fisher function depends on how $H$ mediate amplitudes between the basis states of $A$. The same can be done using the basis of energy eigenvectors. This leads to a clear dependence of Fisher on the off diagonal elements $A_{\alpha,\beta}$. The idea now is to see assuming some properties on the distribution of the energy eigenvalues what is the characteristic value of the Fisher function when you impose the ETH/ergodicity conditions. The suggestion is that in this case the quantum Fisher function goes like the Stam's gaussian bound of classical Fisher.

I dont claim that a rigorous proof of this conjecture is presented in the paper. See more comments later.

Even with all the assumptions, it is not clear to me how Eq.(20) is derived. Where do exp(N) in the denominator comes from ? What is N ? I did not find it anywhere in the text.

Now I realize that $N$ is not clearly defined in the paper. Is simply the number of degrees of freedom with the dimension of Hilbert space $D= e^N$.

The claim below Eq.(20) is based on a statistical argument that is valid for the classical Fisher information, but it is used here for the quantum Fisher ifnormation. This is not so straightforward and certainly more explanation is needed. My understanding is that a specific measurement observable “with gaussian probability distribution” is used here, then the classical Fisher is $1/\sigma^2$ with respect to that measurement observable (which is a specific one). With this, I understand Eq.(14). However, it is not the absolute minimum: it is not guaranteed that I can choose a different observable such that the classical Fisher information is smaller than $1/\sigma^2$. Maybe the author is considering a natural observable, but explanation is needed.

As explained above the target of the paper is to see if the ETH conditions for a pair $A,H$ can be associated with some special property of the associated quantum Fisher function $F(A,H)$. The observation and suggestion is that under these conditions the {\it quantum} $F(A,H)$ saturates Stam's inequality for a {\it classical} Fisher function. Certainly Stam's inequality is valid only for classical Fisher. My personal interpretation, that can be completely wrong, but that explains the additional comments in the paper is that what this result indicates is that the uncertainty in time defined by quantum Cramer Rao theorem is in the ETH/ergodicity conditions of the order of complexity time. In the rest of the comments what I try is to see how the picture changes when we introduce macroscopic observables i.e. $d_{\nu}$ different from one.

In conclusion, I got very interested by the paper and I read it carefully. Unfortunately the paper is not clear and detailed. I got very confused and I did not learned much. As it is, the manuscript is not accessible to any reader. I cannot recommend the current version even though I will be willing to reconsider a more detailed and expanded version, but I will reject a version with incremental improvements.

The reason of the former comments is due to the last sentence of the referee. I can happily try a more detailed and expanded version on the basis of the comments presented above but I am afraid it could be thought as a version with {\it incremental improvements}. Thus I will appreciate very much to know the reaction of the referee to the comments made in this answer before working out an expanded version.

With my best wishes,

Cesar Gomez

---

## Editorial Decision

awaiting_resubmission